# Impact of High-Pressure Homogenization Parameters on Physicochemical Characteristics, Bioactive Compounds Content, and Antioxidant Capacity of Blackcurrant Juice

**DOI:** 10.3390/molecules26061802

**Published:** 2021-03-23

**Authors:** Bartosz Kruszewski, Katarzyna Zawada, Piotr Karpiński

**Affiliations:** 1Department of Food Technology and Assessment, Institute of Food Sciences, Warsaw University of Life Sciences-SGGW, Nowoursynowska 159 C, 02-776 Warsaw, Poland; 2Department of Physical Chemistry, Faculty of Pharmacy with the Laboratory Medicine Division, Medical University of Warsaw, Banacha 1, 02-097 Warsaw, Poland; kzawada@wum.edu.pl; 3Faculty of Computer Science and Food Science, Lomza State University of Applied Sciences (PWSIiP), Akademicka 14, 18-400 Łomża, Poland; pkarpinski@pwsip.edu.pl

**Keywords:** high-pressure homogenization, blackcurrant juice, anthocyanins, colour, antioxidant activity, vitamin C, turbidity

## Abstract

High-pressure homogenization (HPH) is one of the food-processing methods being tested for use in food preservation as an alternative to pasteurization. The effects of the HPH process on food can vary depending on the process parameters used and product characteristics. The study aimed to investigate the effect of pressure, the number of passes, and the inlet temperature of HPH processing on the quality of cloudy blackcurrant juice as an example of food rich in bioactive compounds. For this purpose, the HPH treatment (pressure of 50, 150, and 220 MPa; one, three, and five passes; inlet temperature at 4 and 20 °C) and the pasteurization of the juice were performed. Titratable acidity, pH, turbidity, anthocyanin, vitamin C, and total phenolics content, as well as colour, and antioxidant activity were measured. Heat treatment significantly decreased the quality of the juice. For processing of the juice, the best were the combinations of the following: one pass, the inlet temperature of 4 °C, any of the used pressures (50, 150, and 220 MPa); and one pass, the inlet temperature of 20 °C, and the pressure of 150 MPa. Vitamin C and anthocyanin degradation have been reported during the HPH. The multiple passes of the juice through the machine were only beneficial in increasing the antioxidant capacity but negatively affected the colour stability.

## 1. Introduction

Blackcurrants are widely grown in a temperate climate in Europe, Asia, North America, Australia, and New Zealand. According to the International Blackcurrant Association, Poland is the world leader in the production of blackcurrant: more than half of the volume of the berries is collected in this country. Due to the high content of biologically active components, blackcurrants are a valuable raw material in the production of juices, nectars, jams, dried foods, liqueurs, and wines. Blackcurrant fruits are rich in health-promoting bioactive substances such as fibre, vitamin C, and polyphenols (e.g., anthocyanins, phenolic acids, flavanols, flavonols, and proanthocyanidins) [1]. However, the qualitative and quantitative composition of blackcurrant bioactive compounds has been revealed to vary among cultivars and growing years [2,3]. Furthermore, the processing operations significantly affect the final composition profile of blackcurrant products, e.g., juices [4,5].

High-pressure homogenization (HPH) is considered the emerging technology in the processing and preservation of liquid food products. This technology relies on the pressing (50–400 MPa) of the product through a narrow gap of the disruption valve which simultaneously inflicts various physical phenomena such as cavitation, turbulence, friction, and shear forces [6]. The result of the process is primarily a reduction in the particle size and mechanical disintegration of the microorganisms [7]. Previous studies have shown varying influence of the HPH on the profile of bioactive compounds of the processed food [8,9,10]. Generally, the application of the HPH allows for retaining a higher amount of bioactive components than classic pasteurization [11,12]. This gives the opportunity of using the HPH for the production of minimally processed foods with high nutritional value and more than 40 days of shelf life under refrigerated storage [13].

According to the literature discussing the HPH processing influence on the quality of vegetable/fruit juices and purees, it was shown that each case should be considered separately according to the type of raw material and the HPH device used [8,14,15]. To the best of our knowledge, there is no information related to the qualitative and nutritional changes during the HPH processing of any blackcurrant food product.

Therefore, in this study, we aimed to investigate the effect of pressure, the number of passes, and the inlet temperature of the HPH processing on the physicochemical characteristics, colour stability, antioxidant capacity, and the content of selected bioactive compounds in a highly valuable product such as blackcurrant juice. In addition, the HPH and the thermal treatment of juice were compared.

## 2. Results and Discussion

### 2.1. Changes in Physicochemical Characteristics of the Juice

According to the results (Table 1), blackcurrant juice had a very low pH, which is characteristic of the Tiben variety. Pasteurization treatment did not affect the pH of the juice. For some combinations of HPH parameters, there was a slight, nonsignificant decrease in the pH of samples, probably because of the extraction of organic acids from residual particles. Neither the pasteurization nor the HPH treatment caused any significant change in the titratable acidity (TA) value of the blackcurrant juice. The effect of the inlet temperature, the number of passes, or the level of pressure has not been demonstrated either. It is in contrast to earlier studies of different fruit juices. Previously, Velázquez-Estrada et al. [16] reported that TA of orange juice was more affected by the inlet temperature than the pressure level. Yildiz [12] found a decrease in pH and no effect on TA through either the thermal or the HPH treatment of peach juice.

Obtained results demonstrated that total soluble solids (TSS) content decreases during thermal pasteurization, probably due to the reduction in the quantity of reducing sugars and denaturation of residual proteins [16,17,18]. However, Suárez-Jacobo et al. [19] did not find any remarkable changes in the TSS after 4 min of pasteurization of apple juice. These differences might be due to the different matrix composition and heat doses delivered to the juices. Samples homogenized at the inlet temperature of 20 °C showed a TSS value of more than 16 °Brix. The pressure of 150 MPa at both inlet temperatures was optimal for getting juice with the highest TSS amount—about 0.6 °Brix higher than the untreated sample. Increasing the pressure from 150 to 220 MPa resulted in a TSS content decrease, due to the aggregation and precipitation of part of soluble substances that was caused by more severe processing conditions, e.g. the elevated temperature in the valve, higher shear forces, and cavitation. Increasing the number of passes did not affect significantly the TSS content. Other authors observed the impact of pressure and the number of passes on TSS [9,17] or reported no effect of the HPH treatment on this parameter in fruit juices [10,19,20].

### 2.2. Nephelometric Turbidity of Juice

The pasteurized and the HPH processed samples exhibited much lower turbidity than the control sample, and this effect was statistically significant (Figure 1). Pasteurized blackcurrant juice was characterized by a turbidity of about 70 NTU. Under the influence of heat suspended particles precipitate and settle down by gravity. The precipitation was visibly noticeable.

The increase in both the pressure and the number of passes during the HPH treatment significantly reduced the turbidity of samples. The best results were achieved by homogenizing the juice through the application of five passes regardless of the inlet temperature and pressure (21–53 NTU). The pressure was more important only when one or three passes were applied. The inlet temperature seems to have no effect on the turbidity. The decrease in the NTU of the HPH processed samples can be explained by the reduction in the size of particles as other authors proved [10,21]. Smaller particles allow more light to go through the sample without reflecting it. Some authors have pointed to the positive correlation between the particle size and serum cloudiness of the HPH juices [22].

### 2.3. Juice Colour Stability

Both HPH processing and pasteurization affected the colour of juice (Table 1, Appendix A). The pasteurization process resulted in the greatest change in colour (ΔE* = 4.26), which was visually observable. This change was indicated by an increase in the lightness of the sample, a decrease in a* value, which corresponds to decreased anthocyanin content of juice, and an increase in b* value, probably due to nonenzymatic browning. The increase in L* value for the pasteurized juice can be attributed to partial precipitation of unstable, suspended particles present in the cloudy blackcurrant juice, induced by heat. In addition, other authors noted similar observation for pasteurized grapefruit juice [23].

The HPH process conditions caused mostly minor changes in juice colour (ΔE* values in range 0.49–3.33), which were smaller than pasteurization did. Increasing the HPH number of passes contributed to the enhancement of L* value and decrease in a* value. Juice lightness increased probably due to the fragmentation of suspended particles which manifested through the size reduction and shape change. However, at the inlet temperature of 20 °C and all three pressure values, some precipitated particles were noticed at the bottom of the HPH juice containers, which, as in the case of pasteurization, could explain further increase in L* value. Our homogenizer was not equipped with a second low-pressure valve preventing or limiting the reaggregation phenomena [24], and reaggregation and coalescence of particles occurred. The decrease in juice redness can be explained by the partial loss of main pigment constituents. Homogenization slightly increased the b* value of the measured juice colour. This means that even a short exposure of the juice to elevated temperature could initiate nonenzymatic browning reactions that result in increase in yellow and brown pigments content.

In the literature, the HPH processing has variously impacted the colour of food. Homogenization of smoothie resulted in the increase in L* value and a* value, and an adverse change in hue angle parameter, while the homogenization of strawberry juice made it darker and redder, with b* value increased [17,25]. However, it is necessary to consider the effect of the device design and ingredients characteristics on the final colour of the obtained product.

There were statistically significant (*p* < 0.05) correlations between ΔE* values of the HPH juice and the number of passes, as well as the inlet temperature. Increasing both homogenization parameters resulted in ΔE* values increase. Out of 18 HPH parameters combinations, four samples, including only one prepared at the inlet temperature of 4 °C, have shown noticeable colour change (ΔE* > 2.0). The most severe HPH conditions in our experiment (220 MPa, 20 °C, three and five passes) induced the ΔE* value to exceed 3.0, which was a substantial change in colour.

### 2.4. Changes in Anthocyanins Content

Four anthocyanin monomers have been quantified (Table 2). The primary anthocyanin in blackcurrant cloudy juice was delphinidin-3-*O*-rutinoside, comprising 49.3% of all anthocyanins in the control sample.

The pasteurization of the juice caused a meaningful decrease in the anthocyanin content up to 78% of the basic value. The heat treatment did not change the proportion of individual monomers.

The great majority of HPH parameters combinations reduced the total anthocyanin content of the juice (Figure 2). However, the level of reduction varied considerably (2.3–24%). An increase in the pressure or the number of passes alone did not affect the anthocyanin quantity. Only at the inlet temperature of 20 °C, the increase in the pressure and number of passes affected anthocyanin content. All combinations of the HPH parameters with an inlet temperature of 4 °C allowed anthocyanin monomers to be preserved with an average retention of 91.2%. All combinations of the HPH parameters with an inlet temperature of 20 °C decreased the anthocyanin content a little bit more (to an average of 87.3%). The homogenization parameters that induced the highest degree of anthocyanin degradation were 220 MPa/one, three, and five passes/20 °C, where anthocyanin sum was near the level obtained after 1 min of pasteurization. None of the HPH parameters changed the distribution of individual monomers (observed deviation ± 1.5%).

We presume that the mechanism of anthocyanins degradation during the pasteurization and HPH processing of the blackcurrant juice was completely different. Anthocyanins are susceptible to thermal degradation, which is the main mechanism during pasteurization, and not HPH treatment, although the effect of heat generated at the HPH valve could not be excluded despite cooling after each passing used in the experiment. In contrast, under the HPH treatment, degradation of anthocyanins was probably due to oxidation reactions with the participation of oxygen brought to the juice by the turbulent process of homogenization. These reactions were counteracted, to some extent, by reducing substances such as ascorbic acid and phenolic acids, which are abundant in blackcurrant berries [1]. We assume that the impact of mechanical stress on the stability of anthocyanin molecules was minimal, as other authors have proven [26]. Anthocyanins released from different kinds of complexes in cloudy juice, due to the reduction in the particle size should also be taken into account [25].

### 2.5. Total Vitamin C, Ascorbic Acid, and Dehydroascorbic Acid Stability

One-minute pasteurization reduced the total vitamin C content of blackcurrant cloudy juice by about 30% and the ascorbic acid (AA) concentration by approximately half of the original quantity (Table 3). The dominant degradation of AA was due to the degradation mechanism, as AA is transformed to dehydroascorbic acid (DHAA) form, and then DHAA undergoes further reactions like hydrolysis to 2,3-diketo-L-gulonate or oxidation to a range of products such as cyclic oxalyl threonate, oxalyl threonate, L-threonic acid, and oxalic acid [27].

In 9 out of 18 variants of HPH parameters, the total vitamin C content was less than or equal to the value obtained for the pasteurized samples. At the inlet temperature of 4 °C, the applied pressure has not influenced the AA, DHAA, and total vitamin C concentration. An important factor, in this case, was the number of times the juice passed through the homogenizer. Cloudy juice samples showed similar retention of both forms of vitamin C within the pass number. At one, three, and five cycles the total vitamin C oscillated around 94.5%, 82.7%, and 70.7% of the original value, respectively. The lack of pressure effect can be linked to the lack of valve temperature effect on the tested juices. However, low (4 °C) inlet temperature of the juice must be taken into account. At the inlet temperature of 20 °C, increasing both pressure and the number of passes resulted in the progressive decrease in AA and DHAA content. This was very evident for samples homogenised at 220 MPa. All five passes at 20 °C resulted in obtaining total vitamin C concentration below the level of the pasteurized sample.

The AA is very unstable and sensitive to temperature, and despite a very short residence time lasting a few seconds (longer at higher pressures) in the valve of the machine, it may have undergone a degradation process before the juice was cooled. Higher pressures created stronger mechanical forces and therefore generated higher temperatures on the valve [6,7,24]. However, as the results described above show, this situation occurs only at the inlet temperature of 20 °C. The results from the HPH treatment indicated that the increasing number of passes contributed to more intensive oxidation of both forms of vitamin C. The reason could be that during homogenization the juice gets aerated and mechanical forces favour the process of oxidation. The samples were cooled in a blast freezer in tightly sealed bottles, meaning accidental aeration during the cooling process between each pass can be excluded.

Some authors indicated that seals in an HPH device, as well as the valve design and physicochemical characteristics of the homogenized fluid, strongly affect the fate of bioactive compounds [7,11,24]. In this regard, we tested the HPH homogenizer by pumping the juice at minimal pressure of 20 MPa at the inlet temperature of 12 °C and made proper determinations. The experiment has confirmed that the construction of this homogenizer causes at a single pass a loss of about 4–5% of total vitamin C content. This information may also partly explain the losses of other bioactive substances of the tested juice. However, it was difficult to determine whether this was due to seals, valve design or the characteristics of the metal surface inside. According to the literature, DHAA can be hydrolysed especially in highly acidic solutions [27,28]. Our blackcurrant juice samples have the pH of about 2.5, which may have favoured this type of reaction. Other authors reported various findings on the impact of HPH on vitamin C content, depending on the parameters and the type of product. Yu et al. [26] noted retention of only 58.7% and 35.3% of AA at 200 MPa after one and three passes of mulberry juice, respectively. Tribst et al. [29] homogenized mango nectar at 200 and 300 MPa with a single pass which resulted in roughly 50% losses of vitamin C. However, Guan et al. [30] found no significant changes in the ascorbic acid amount after one and three passes, at 190 MPa, and 60 °C of inlet temperature in mango juice. In another study, Benjamin and Gamrasni [20] observed no loss of AA at 100 MPa but noted a statistical loss of AA at 150 MPa in pomegranate juice.

### 2.6. Total Phenolic Content and Antioxidant Capacity

One minute of pasteurization reduced the content of total phenolics (TPC) by 26.6% due to the thermal degradation of compounds (Figure 3). HPH processing caused a slight change in the TPC of cloudy blackcurrant juice by about ± 5% of the original value. Only the samples homogenized at 220 MPa, three and five passes, at the inlet temperature of 20 °C exhibited the TPC value at about 86%, which was still more than the TPC content in the pasteurized sample.

The results suggest that the HPH processed juices despite losses in anthocyanins and vitamin C may have the TPC value close to the crude juice. We believe that it did not necessarily imply high retention of Folin–Ciocalteu reagent-reactive components. This phenomenon can be explained by the formation of derivatives of already present phenolic compounds and/or new ones through the hydrolysis and depolymerization of complexes induced directly by high mechanical forces during the HPH process, as well as the enhanced extraction of these compounds from the mechanically destroyed cells [8,31]. It is important to remember that in this research we treated a pH 2.5 cloudy blackcurrant juice. Similar to our study, Karacam et al. [17] determined a higher content of total phenolics in strawberry juice after homogenization at 100 MPa and at least two passes, in comparison to the control sample. Saricaoglu et al. [9] found a significantly lower content of polyphenols after the HPH was done at 75 MPa on rosehip juice, whereas the application of 125 MPa had no effect.

Heat treatment decreased the antioxidant capacity of the juice measured by both DPPH-EPR and ORAC assay (Table 4). The HPH processing with the inlet temperature of 4 °C reduced the antioxidant activity of juice by 6.8 up to 14.7% (DPPH-EPR assay), and by 1.0 up to 13.0% (ORAC assay) of the original value. The ORAC test showed a higher rate of antioxidant activity preservation in the HPH juice samples (only at the inlet temperature of 4 °C) in comparison to the DPPH-EPR test. Homogenization with the inlet temperature of 20 °C in the vast majority of variants (50 and 150 MPa with various numbers of passes) facilitated an increase in the antioxidant capacity of the juice. Additionally, increasing the number of passes resulted in higher antioxidant values. However, the increase in the antioxidant capacity was found to be greater via the DPPH-EPR than the ORAC method. Discrepancies between methods used for the antioxidant capacity determination are because of the difference in the temperature, mechanisms, and nature of reactions utilized for measurement, as well as due to the complexity of the molecular structure of the antioxidants [19,32].

Despite the determined losses in anthocyanins or vitamin C, the antioxidant activity of the juice may have increased because the degradation products and their derivatives also exhibit antioxidant activity, as many authors demonstrated in the literature [27,33,34,35]. The usage of 220 MPa especially with the application of three and five passes decreased the antioxidant capacity of juice. Different physical phenomena such as cavitation, turbulence, friction, and shear forces must have degraded much of the substances responsible for the antioxidant activity, which coincides with higher losses of anthocyanins and vitamin C among the tested samples. Perhaps at these parameters, the multiple short-term exposure to the highest recorded temperature (62 °C) in the homogenizer valve affected the direction of the reactions and the chemical properties of the resulting compounds.

This study showed that the inlet temperature of the juice together with the pressure of homogenization were both important in terms of maintaining or even increasing its antioxidant activity. Still, different authors dealing with the HPH processing of food products have shown varying results, which indicates the comprehensiveness of the subject. Suárez-Jacobo et al. [19] did not find any significant differences in the antioxidant capacity between raw and HPH-treated apple juice using the DPPH method. On the other hand, when the ORAC test was used, they noticed that increasing pressure increases antioxidant capacity, with the inlet temperature not playing any significant role. However, Yu et al. [26] reported a significant decrease in ORAC values of mulberry juice, attributing the losses to the degradation of anthocyanins and phenolic acids via oxidation. Karacam et al. [17] indicated a substantial role of inactivation of native enzymes in preventing the decrease in antioxidant capacity of the processed juices. Thus, it seems that the effect of the matrix is important for the effect of HPH treatment on antioxidant properties.

### 2.7. Principal Component Analysis of Gathered Data

Principal component analysis (PCA) graphs, based on the first two principal components which explained 79.4% of the total variance, show the similarity of the processed samples towards raw or pasteurized juice (Figure 4). Based on the PCA graphs, the best parameters of HPH treatment for preserving the quality of the cloudy blackcurrant juice was the combination of one pass, the inlet temperature of 4 °C, any of the used pressures (50, 150, and 220 MPa); and one pass, the inlet temperature of 20 °C, 150 MPa. The combination of the parameters 50 or 150 MPa, three or five passes, and the inlet temperature of 20 °C gave superior enhancement of the antioxidant capacity of juice but at the cost of big losses of vitamin C, some anthocyanins, and, in the case of five passes, worse colour stability. Due to the highest similarity to the pasteurized juice, 220 MPa with 3 and 5 passes at the inlet temperature of 20 °C is not recommended for utilization.

## 3. Materials and Methods

### 3.1. Blackcurrant Berries and Juice Production

The frozen blackcurrant berries (*Ribes nigrum* L.) packed in polyethene bags which were kept in polystyrene containers were bought from a professional supplier in central Poland. The Tiben variety was selected as it is characterized by a high-processing value and is frequently used by the food-processing industry for the production of blackcurrant juices, nectars, and juice concentrates. The Tiben variety is a cross between two other varieties: Titania and Ben Nevis, and it originates from Poland.

Loose frozen fruits were thawed in a stainless-steel beaker in a water bath, then quickly heated to 80 °C, and blanched at this temperature with mechanical stirring for 3 min in order to inactivate the native enzymes. Afterwards, blackcurrants were quickly cooled to 55 °C in an ice bath and were comminuted by a hand blender. Then, the optimal dosage of pectolytic enzymes worked for 2 h at 55 °C to increase the juice yield and to decrease the viscosity of the pulp. The juice was pressed on the self-made laboratory juice press (yield 70%) equipped with a filter cloth with a mesh opening of 500 μm. The obtained cloudy juice was immediately cooled to 4 °C in an ice bath and divided into three batches: HPH processed, pasteurized, and control (unprocessed).

### 3.2. HPH Treatment and Thermal Pasteurization of the Juice

HPH processing was conducted using a high-pressure homogenizer IKA 2000/4-SH5 (IKA^®^-Werke GmbH & Co. KG, Straufen, Germany) by combining the following conditions: pressure: 50, 150, and 220 MPa; the number of sample passes through homogenizer: one, three, and five; and the inlet temperature (IT) 4 °C or 20 °C. Prior to the processing, the device was cleaned with 70% ethanol. After each pass, the processed juice was rapidly cooled to the specific inlet temperature in sterile, closed glass bottles in the Blast Chiller MF 30.2 (Irinox S.p.A., Via Madonna di Loreto, Italy). Cooling was necessary to limit the impact of the heat on blackcurrant juice after each pass. The maximum temperature monitored directly after the valve head during the processing was 37, 46, and 62 °C for the 50, 150, and 220 MPa, respectively.

Simultaneously, the other batch of the juice was pasteurized in a water bath LW8M100 (WSL Sp. z.o.o., Świętochłowice, Poland) at 90 °C for 1 min, as other authors did [15,16,30]. The temperature was measured for every 10 s by an electronic recorder MPI-CL-16-4 (Metronic AKP s.c., Kraków, Poland) equipped with thermocouples introduced into additional, filled glass bottles.

### 3.3. Chemicals and Reagents

pH buffer solutions, hydrochloric acid, 0.1 M sodium hydroxide, anhydrous sodium carbonate, and Folin–Ciocalteu reagent were all analytical grade and purchased from Chempur (Piekary Śląskie, Poland). Disodium fluorescein of reagent grade, m-phosphoric acid, acetonitrile, and formic acid of HPLC gradient grade were purchased from Avantor Performance Materials Poland S.A. (Gliwice, Poland). Gallic acid anhydrous, phosphate-buffered saline (PBS), 2,2′-azobis(2-methylpropionamidine) dihydrochloride (AAPH), (±)-6-hydroxy-2,5,7,8-tetramethylchromane-2-carboxylic acid (Trolox), methanol, and 2,2-diphenyl-1-picrylhydrazyl (DPPH) were reagent or analytical grade, purchased from Sigma-Aldrich Sp. z.o.o (Poznań, Poland). Pure water (resistivity of 18.0 MΩ·cm) was also used in the research.

### 3.4. Determination of pH, Titratable Acidity, and Total Soluble Solids

pH and titratable acidity (TA) of blackcurrant juice were analysed by HI 221 pH meter (Hanna Instruments, Olsztyn, Poland). Before analyses, the pH meter was calibrated with buffer solutions, and temperature of juice samples was adjusted to 23 °C.

Titratable acidity was determined by titrating blackcurrant juice to pH 8.1 using 0.1 M sodium hydroxide. Results were expressed as g of citric acid per 100 mL of juice.

Total soluble solids (TSS) were determined with a portable refractometer Refracto 30PX (Mettler-Toledo, Columbus, OH, USA) at 20 °C and were reported as °Brix.

### 3.5. Colour Analysis

The colour of blackcurrant juices was measured with a colourimeter Konica Minolta CM-3600d (Tokyo, Japan) in CIE L*a*b* scale, calibrated against black and white plate standards. The parameters of the device in the transmission mode were set as follows: a standard observer of 10°, and an illuminant D65. L* (lightness), a* (red to green), and b* (yellow to blue) values were measured using a glass cuvette (0.2 cm path) at room temperature. The total colour difference (ΔE*) between untreated and treated juices was calculated by applying Equation (1).
(1)$${\Delta E}^{*}=\;\sqrt{{\left({\Delta L}^{*}\right)}^{2}+{\left({\Delta a}^{*}\right)}^{2}+{\left({\Delta b}^{*}\right)}^{2}}$$

### 3.6. Nephelometric Turbidity Measurement

Direct turbidity expressed as nephelometric turbidity units (NTU) of all blackcurrant juices was managed by a turbidimeter Ratio/XR 43900 model (Hach Lange GmbH, Düsseldorf, Germany) using 15 mL glass vial. The temperature of the juice samples was adjusted to 23 °C before measurement. The device was calibrated against formazin turbidity standards.

### 3.7. Determination of Total Phenolic Content

Total phenolic content (TPC) of blackcurrant juices was determined by using Folin–Ciocalteu reagent according to the well-known method with slight modifications [36]. In brief, 1 mL of juice was dissolved in 25 mL of redistilled water. Then, 0.2 mL of juice solution was mixed with 0.4 mL of Folin–Ciocalteu reagent, 4 mL of redistilled water, and 2 mL of 15% sodium carbonate. After 1 h of blend incubation in a dark place at the temperature of 25 °C, the absorbance measurements were taken with a UV-Vis spectrophotometer at the wavelength of 765 nm (25 °C) against mixed reagents. Total phenolic content was expressed as mg gallic acid equivalents (GAE) per 100 mL of juice, based on the prepared calibration curve.

### 3.8. Determination of Total Vitamin C, AA, and DHAA Content

Total vitamin C, L-ascorbic acid (AA), and L-dehydroascorbic acid (DHAA) were determined by HPLC using the method previously stated in the literature [37]. One mL of blackcurrant juice was transferred to a volumetric flask and filled to 10 mL with 0.1% of m-phosphoric acid. To analyse AA, part of the mixture was filtered through a 0.45 µm PTFE syringe filter to a chromatographic vial. In order to determine total vitamin C, a reducing agent (0.1% dl-dithiothreitol in 0.1% *m*-phosphoric acid) was added into the second part of the mixture (50:50 *v*/*v*) and was left in a dark place for 1 h to reduce DHAA to AA. This reaction solution was also filtered through a 0.45 µm PTFE syringe filter to a chromatographic vial.

The sample volume of 20 µL from each chromatographic vial was injected into the HPLC system with a UV-Vis detector (Shimadzu, Kyoto, Japan). The stationary phase was an Onyx Monolithic C18 column (100 × 4.6 mm, 5 µm, Phenomenex, Torrance, CA, USA). The mobile phase was 0.1% *m*-phosphoric acid solution in pure water. The analysis was carried out using isocratic conditions (1 mL/min), at the column temperature of 25 °C, with detection performed at 254 nm. The calibration curves of the external standard solutions of AA were used. The DHAA content in samples was calculated as the difference between the total vitamin C and AA contents [38]. All results were expressed as mg/100 mL of juice.

### 3.9. Determination of Anthocyanin Content

Anthocyanins analysis was conducted according to the method reported in the literature [38]. Two mL of juice sample was transferred to a volumetric flask and filled to 10 mL with 0.1% aqueous hydrochloric acid. This mixture was filtered on a 0.45 µm PTFE syringe filter to a chromatographic vial.

Analyses were carried out by an HPLC with a UV-Vis detector (Shimadzu, Kyoto, Japan). Separation of anthocyanins was done on C18(2) Luna column (100 Å, 5 µm, 250 × 4.6 mm) from Phenomenex (Torrance, CA, USA). The isocratic elution at flow rate 1 mL/min was performed using a mobile phase of water/acetonitrile/formic acid (83:7:10 *v*/*v*/*v*) at the column temperature of 25 °C. The detector was operated at a wavelength of 520 nm. Monomers of the anthocyanins were identified by comparing their retention times with the retention times library of selected anthocyanin standards. The content of each monomer was calculated based on calibration curve for cyanidin-3-*O*-glucoside standard solutions and expressed as mg per 100 mL of juice.

### 3.10. DPPH-EPR Radical Scavenging Assay

The DPPH radical scavenging test was used according to the method mentioned previously in the literature [39]. Radical scavenging activity was measured by electron paramagnetic resonance (EPR) spectroscopy, in order to eliminate matrix interference in coloured and cloudy juice samples. The samples were diluted 20-fold with methanol. Fifty µL of diluted sample and 35 µL of DPPH methanolic solution (1.3 mM) were mixed, and after 30 min, EPR spectra were measured. The intensity of registered EPR spectra was compared with the blank sample (redistilled water added to the methanolic DPPH solution). EPR spectra were registered at 25 °C using a MiniScope MS200 EPR spectrometer (Magnettech GmbH, Berlin, Germany) operating at the X-band (9.4 GHz), centre field 330.2 mT, sweep width 8.5 mT, sweep time 20 s, modulation amplitude 0.08 mT, and microwave power 6.3 mW. The results were expressed as Trolox equivalents (TE) in µmol TE per L of juice.

### 3.11. ORAC Assay

The oxygen radical absorbance capacity (ORAC) was measured based on the method proposed in the literature [40]. All solutions were prepared in PBS (100 mM, pH 7.4) on the day of measurement. In brief, 30 µL of the juice diluted 1000-fold, and Trolox solution was mixed with 180 µL of fluorescein solution (112 nM) in a 96-well plate and thermostated for 10 min at 37 °C. Then, to each well 100 µL of AAPH solution (100 mM) was added. For each plate, a blank test using 30 µL of PBS was done, and a 5-point Trolox standard curve (10–100 µM) was prepared. The intensity of fluorescence was registered every 70 s for the duration of 90 min at 37 °C with F-7000 Fluorescence Spectrophotometer (Hitachi, Tokyo, Japan), equipped with a microplate reader. Instrument settings were as follows: excitation wavelength 485 nm and emission wavelength of 520 nm. ORAC values were expressed as Trolox equivalents (TE) in µmol TE per L of juice.

### 3.12. Statistics

All data were presented as a mean with standard deviation for three replications of each measurement done for each of two independent repetitions of HPH and thermal treatment experiments. Statistical analyses were conducted using Statistica 13.3 (TIBCO Software Inc., Carlsbad, CA, USA). The effects of the different processing treatment on the measured compounds content were determined by ANOVA analysis of variance. The differences between means were evaluated with the Tukey HSD post hoc test (α = 95%). The gathered quantitative data were used in the principal component analysis (PCA) in order to show the affinity of the processed samples towards raw or pasteurized juice. The results of the determinations performed were qualified for PCA analysis based on a correlation score with the first or second principal component of at least 0.6.

## 4. Conclusions

In this research, the processing of the juice by HPH was compared with pasteurization in order to evaluate the effects of treatment on the physicochemical characteristics, colour stability, bioactive compounds content, and the antioxidant capacity of the cloudy blackcurrant juice. A 1 min pasteurization significantly reduced (*p* < 0.05) the content of anthocyanins and total vitamin C, as well as decreased TSS value, nephelometric turbidity, colour stability, TPC content, and antioxidant capacity but did not affect the pH and TA of juice. The innovative technology of HPH processing affected the quality of the juice depending on the parameters applied. The best combination of HPH treatment in preserving the quality of the juice was the application of one pass, inlet temperature of 4 °C, and any of the used pressures (50, 150, and 220 MPa); and one pass, the inlet temperature of 20 °C, and 150 MPa. The combination of parameters: 50 or 150 MPa, three or five passes, and the inlet temperature of 20 °C gave superior enhancement of the antioxidant capacity of juice but at the cost of big losses in vitamin C, some anthocyanins and, in the case of five passes, worse colour stability. The inlet temperature of 20 °C together with the pressure of 220 MPa was unfavourable for the bioactive components and the colour stability of the juice. Together with the increase in pressure and number of passes, the nephelometric turbidity was greatly reduced. All the HPH treatments caused fewer changes in juice colour than 1 min pasteurization.

In general, the high-pressure homogenization has a great potential to preserve or even improve the bioactive and physicochemical quality of blackcurrant fruit juice, but adequate parameters should be considered depending on the desired final characteristics of the product.

## Figures and Tables

**Figure 1 molecules-26-01802-f001:**
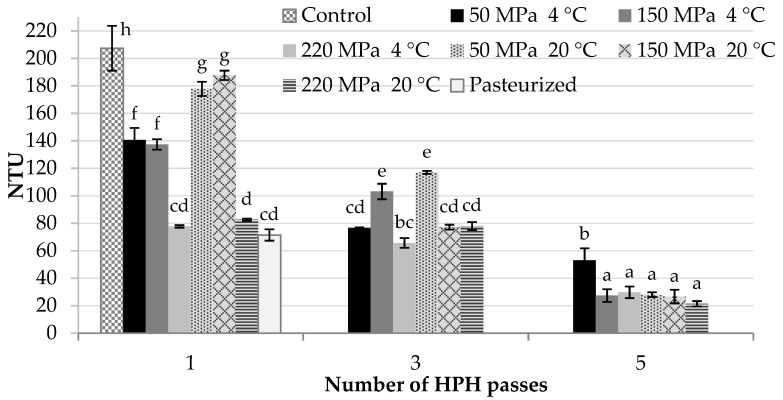
Nephelometric turbidity of cloudy blackcurrant juice treated with high-pressure homogenization, or pasteurization. The values with different superscript letters are significantly different (*p* < 0.05).

**Figure 2 molecules-26-01802-f002:**
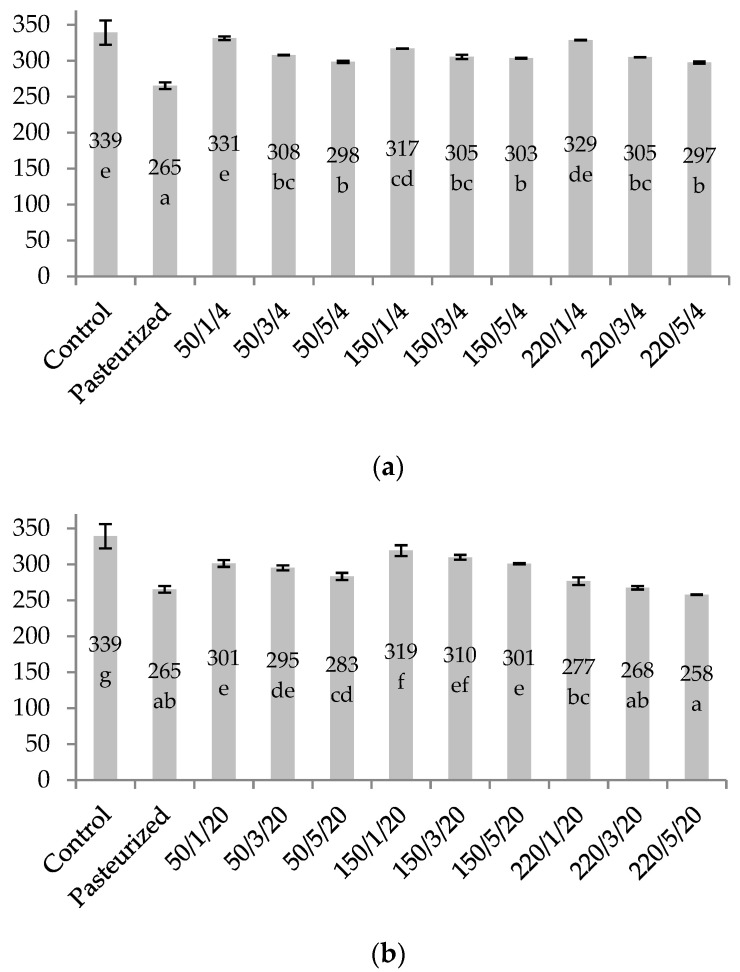
The total concentration of anthocyanins (mg/100 mL) determined in cloudy blackcurrant juice treated with high-pressure homogenization or pasteurization. Samples were labelled as level of pressure/number of passes/inlet temperature. (**a**) HPH samples with the inlet temperature of 4 °C and (**b**) HPH samples with the inlet temperature of 20 °C. The values with different superscript letters (a–g) are significantly different (*p* < 0.05). Statistics were performed separately for (**a**) and (**b**).

**Figure 3 molecules-26-01802-f003:**
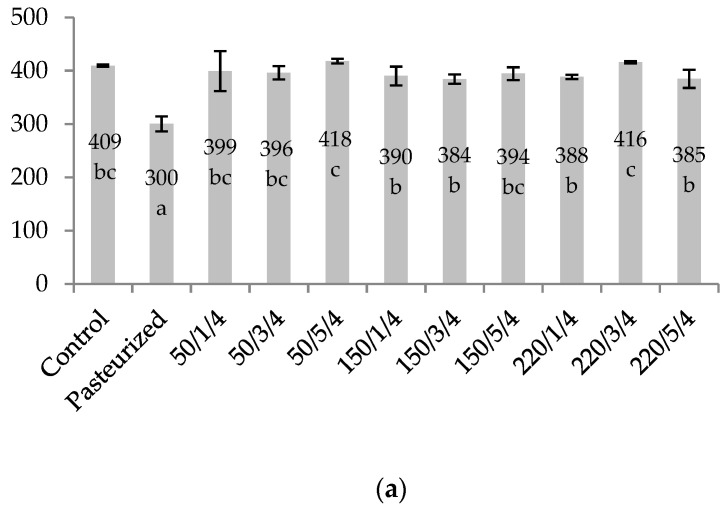
Total phenolic content (mg GAE/100 mL) determined in cloudy blackcurrant juice treated with high-pressure homogenization or pasteurization. Samples were labelled as level of pressure/number of passes/inlet temperature. (**a**) HPH samples with the inlet temperature of 4 °C; (**b**) HPH samples with the inlet temperature of 20 °C. The values with different superscript letters (a–e) are significantly different (*p* < 0.05). Statistics were performed separately for figure (**a**) and figure (**b**).

**Figure 4 molecules-26-01802-f004:**
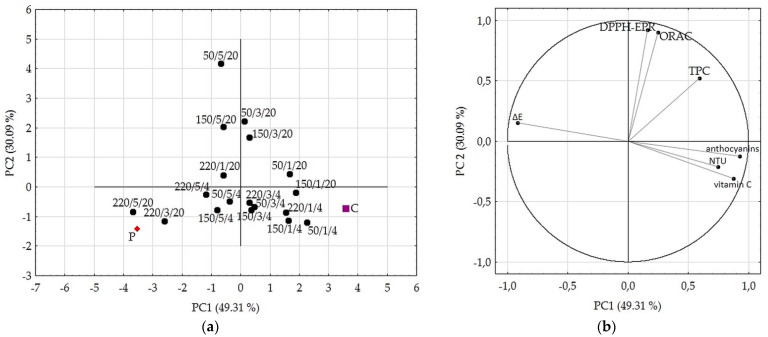
Principal component analysis (PCA) analyses results. (**a**) Score plot, PC1 versus PC2 of all samples; (**b**) Score plot, PC1 versus PC2 of data from determinations used as variables. C–control sample/raw juice, P–pasteurized juice, and HPH samples were labelled as level of pressure/number of passes/inlet temperature.

**Table 1 molecules-26-01802-t001:** The physicochemical characteristics: pH, titratable acidity (TA), total soluble solids (TSS), and the total colour difference (ΔE*) of cloudy blackcurrant juice treated with high-pressure homogenization or pasteurization.

Treatment			pH	TA (g of Citric Acid /100 mL)	TSS (Brix)	ΔE*
Control Sample/Raw Juice			2.48 ± 0.04 ^a^	5.05 ± 0.06 ^a^	15.83 ± 0.05 ^bc^	-
Pasteurization 90 °C, 1 min			2.52 ± 0.04 ^a^	4.94 ± 0.06 ^a^	15.32 ± 0.05 ^a^	4.26 ± 0.06 ^m^
Homogenization						
Inlet Temp. °C	Pressure MPa	Number of Passes				
4	50	1	2.45 ± 0.04 ^a^	5.05 ± 0.07 ^a^	15.70 ± 0.09 ^bc^	0.49 ± 0.04 ^a^
		3	2.45 ± 0.05 ^a^	5.05 ± 0.07 ^a^	15.70 ± 0.10 ^bc^	1.38 ± 0.03 ^de^
		5	2.43 ± 0.05 ^a^	5.10 ± 0.04 ^a^	15.90 ± 0.14 ^cdef^	1.78 ± 0.06 ^fg^
	150	1	2.47 ± 0.04 ^a^	5.00 ± 0.03 ^a^	16.45 ± 0.07 ^h^	0.89 ± 0.03 ^bc^
		3	2.50 ± 0.03 ^a^	4.95 ± 0.07 ^a^	16.50 ± 0.05 ^h^	1.42 ± 0.03 ^de^
		5	2.46 ± 0.04 ^a^	4.90 ± 0.05 ^a^	16.40 ± 0.11 ^h^	1.44 ± 0.02 ^e^
	220	1	2.51 ± 0.04 ^a^	5.05 ± 0.04 ^a^	15.75 ± 0.07 ^cd^	0.80 ± 0.06 ^b^
		3	2.49 ± 0.03 ^a^	5.05 ± 0.07 ^a^	15.85 ± 0.15 ^cde^	1.45 ± 0.03 ^e^
		5	2.49 ± 0.02 ^a^	4.95 ± 0.06 ^a^	15.70 ± 0.12 ^bc^	2.32 ± 0.03 ^h^
20	50	1	2.44 ± 0.01 ^a^	5.05 ± 0.07 ^a^	16.20 ± 0.14 ^fg^	1.71 ± 0.03 ^f^
		3	2.43 ± 0.02 ^a^	5.15 ± 0.08 ^a^	16.15 ± 0.06 ^efg^	2.42 ± 0.04 ^h^
		5	2.46 ± 0.03 ^a^	5.10 ± 0.04 ^a^	16.15 ± 0.07 ^efg^	2.70 ± 0.05 ^i^
	150	1	2.44 ± 0.02 ^a^	5.15 ± 0.05 ^a^	16.30 ± 0.08 ^gh^	0.95 ± 0.02 ^c^
		3	2.46 ± 0.04 ^a^	5.10 ± 0.03 ^a^	16.45 ± 0.13 ^h^	1.32 ± 0.02 ^d^
		5	2.50 ± 0.04 ^a^	4.90 ± 0.07 ^a^	16.50 ± 0.10 ^h^	1.89 ± 0.03 ^g^
	220	1	2.46 ± 0.01 ^a^	5.10 ± 0.05 ^a^	16.05 ± 0.13 ^defg^	2.85 ± 0.07 ^j^
		3	2.46 ± 0.05 ^a^	5.10 ± 0.06 ^a^	16.15 ± 0.09 ^efg^	3.11 ± 0.06 ^k^
		5	2.48 ± 0.02 ^a^	4.95 ± 0.07 ^a^	16.25 ± 0.10 ^gh^	3.33 ± 0.06 ^l^

The values in a column with different superscript letters (a–l) are significantly different (*p* < 0.05).

**Table 2 molecules-26-01802-t002:** Anthocyanin monomers content (mg/100 mL) in cloudy blackcurrant juice treated with high-pressure homogenization or pasteurization.

Treatment			Del-3-glu ^1^	Del-3-rut ^2^	Cya-3-glu ^3^	Cya-3-rut ^4^
Control sample/raw juice			64.3 ± 2.6 ^e^	167.1 ± 6.5 ^e^	19.0 ± 1.0 ^e^	88.7 ± 4.4 ^f^
Pasteurization 90 °C, 1 min			48.6 ± 2.8 ^a^	137.2 ± 3.8 ^bc^	12.7 ± 1.0 ^a^	66.8 ± 3.0 ^a^
Homogenization						
Inlet Temp. °C	Pressure MPa	Number of Passes				
4	50	1	66.4 ± 1.4 ^f^	160.6 ± 2.2 ^e^	19.5 ± 0.5 ^e^	84.8 ± 1.6 ^e^
		3	60.1 ± 0.1 ^c^	151.8 ± 1.0 ^cd^	17.5 ± 0.1 ^d^	78.6 ± 1.4 ^c^
		5	58.4 ± 0.2 ^bc^	148.2 ± 1.7 ^c^	16.5 ± 0.5 ^cd^	75.2 ± 0.2 ^b^
	150	1	60.4 ± 0.2 ^c^	155.1 ± 1.6 ^d^	17.7 ± 0.4 ^d^	83.6 ± 2.1 ^d^
		3	62.3 ± 1.3 ^d^	146.5 ± 0.4 ^c^	17.7 ± 1.2 ^d^	78.8 ± 0.9 ^c^
		5	62.8 ± 0.5 ^d^	140.4 ± 0.1 ^bc^	17.8 ± 0.1 ^d^	82.5 ± 0.2 ^d^
	220	1	61.8 ± 1.5 ^d^	168.5 ± 0.7 ^f^	17.0 ± 0.7 ^cd^	81.4 ± 0.3 ^d^
		3	57.4 ± 0.9 ^bc^	155.6 ± 0.6 ^d^	15.4 ± 0.4 ^c^	76.6 ± 0.3 ^c^
		5	54.6 ± 1.2 ^b^	154.8 ± 0.1 ^d^	14.0 ± 1.0 ^c^	73.9 ± 0.6 ^b^
20	50	1	54.8 ± 2.6 ^b^	156.8 ± 0.3 ^d^	15.1 ± 0.8 ^bc^	74.5 ± 1.7 ^b^
		3	55.1 ± 0.5 ^b^	149.9 ± 0.8 ^c^	15.1 ± 0.3 ^bc^	75.1 ± 0.3 ^b^
		5	52.6 ± 1.2 ^a^	142.9 ± 1.6 ^bc^	14.8 ± 1.4 ^bc^	72.8 ± 0.8 ^b^
	150	1	61.0 ± 2.6 ^cd^	161.8 ± 2.2 ^e^	16.5 ± 1.2 ^cd^	79.8 ± 1.4 ^cd^
		3	57.2 ± 1.3 ^bc^	158.9 ± 0.7 ^e^	16.9 ± 0.7 ^cd^	76.7 ± 0.8 ^c^
		5	57.0 ± 0.5 ^b^	153.8 ± 1.2 ^cd^	15.7 ± 0.5 ^c^	74.4 ± 0.3 ^b^
	220	1	53.7 ± 1.5 ^ab^	134.9 ± 0.5 ^ab^	15.7 ± 0.7 ^c^	72.2 ± 2.5 ^b^
		3	51.9 ± 0.5 ^a^	131.9 ± 1.9 ^ab^	15.0 ± 0.1 ^bc^	68.7 ± 0.1 ^a^
		5	53.3 ± 0.1 ^a^	124.1 ± 0.4 ^a^	13.9 ± 0.1 ^ab^	66.4 ± 1.2 ^a^

^1^ Del-3-glu: delphinidin-3-*O*-glucoside; ^2^ Del-3-rut: delphinidin-3-*O*-rutinoside; ^3^ Cya-3-glu: cyanidin-3-*O*-glucoside; and ^4^ Cya-3-rut: cyanidin-3-*O*-rutinoside. The values in a column with different superscript letters (a–f) are significantly different (*p* < 0.05).

**Table 3 molecules-26-01802-t003:** Ascorbic acid, dehydroascorbic acid, and total vitamin C content (mg/100 mL) in cloudy blackcurrant juice samples treated with high-pressure homogenization or pasteurization.

Treatment			Ascorbic Acid	Dehydroascorbic Acid	Total Vitamin C	% Value Relative to Raw Juice
Control sample/raw juice			110.5 ± 4.1 ^h^	68.1 ± 2.6 ^efg^	178.6 ± 4.9 ^g^	100.0
Pasteurization 90 °C, 1 min			57.4 ± 9.2 ^a^	68.2 ± 2.1 ^fg^	125.6 ± 7.1 ^bc^	70.3
Homogenization						
Inlet Temp. °C	Pressure MPa	Number of Passes				
4	50	1	101.2 ± 6.9 ^fg^	67.7 ± 8.5 ^efg^	168.8 ± 1.6 ^fg^	94.5
		3	76.3 ± 4.4 ^cd^	75.2 ± 0.9 ^gh^	151.6 ± 5.3 ^d^	84.9
		5	64.8 ± 3.4 ^ab^	66.7 ± 2.4 ^efg^	131.5 ± 5.8 ^c^	73.6
	150	1	108.8 ± 2.3 ^gh^	69.6 ± 6.9 ^g^	168.4 ± 4.5 ^fg^	94.3
		3	76.0 ± 2.5 ^cd^	81.3 ± 10.2 ^h^	147.4 ± 7.7 ^d^	82.5
		5	64.2 ± 1.0 ^ab^	56.5 ± 0.9 ^bcd^	120.2 ± 1.9 ^bc^	67.6
	220	1	97.7 ± 2.1 ^f^	71.5 ± 6.1 ^gh^	169.27± 4.0 ^fg^	94.7
		3	68.6 ± 1.6 ^bc^	75.5 ± 2.0 ^gh^	144.1 ± 0.4 ^d^	80.7
		5	62.3 ± 0.0 ^ab^	64.3 ± 3.0 ^def^	126.6 ± 3.0 ^bc^	70.9
20	50	1	99.5 ± 0.1 ^f^	66.6 ± 0.2 ^efg^	166.1 ± 0.4 ^ef^	93.0
		3	77.7 ± 4.1 ^d^	51.4 ± 6.7 ^bc^	129.1 ± 10.7 ^c^	72.3
		5	75.0 ± 2.5 ^cd^	45.6 ± 3.3 ^ab^	120.6 ± 5.8 ^bc^	67.5
	150	1	87.7 ± 2.7 ^e^	67.9 ± 1.5 ^efg^	155.6 ± 4.2 ^de^	87.1
		3	70.3 ± 4.0 ^bcd^	57.8 ± 3.2 ^cde^	128.1 ± 7.3 ^c^	71.7
		5	61.6 ± 2.0 ^ab^	53.8 ± 8.3 ^bcd^	115.4 ± 6.3 ^b^	64.6
	220	1	93.0 ± 0.0 ^ef^	57.1 ± 2.0 ^cde^	150.1 ± 2.0 ^d^	84.1
		3	76.4 ± 4.2 ^cd^	39.3 ± 5.4 ^a^	115.6 ± 1.2 ^b^	64.8
		5	59.2 ± 3.5 ^a^	40.4 ± 0.2 ^a^	99.5 ± 3.3 ^a^	55.7

The values in a column with different superscript letters (a–h) are significantly different (*p* < 0.05).

**Table 4 molecules-26-01802-t004:** Antioxidant capacity of cloudy blackcurrant juice samples treated with high-pressure homogenization or pasteurization.

Treatment			DPPH-EPR [µM Tr/L]	% Value Relative to Raw Juice	ORAC [µM Tr/L]	% Value Relative to Raw Juice
Control sample/raw juice			38,128 ± 4229 ^abc^	100.0	72,505 ± 4306 ^def^	100.0
Pasteurization 90 °C, 1 min			33,304 ± 749 ^a^	87.3	65,342 ± 3793 ^bc^	90.1
Homogenization						
Inlet Temp. °C	Pressure MPa	Number of Passes				
4	50	1	33,353 ± 2098 ^a^	87.5	69,633 ± 1775 ^cdef^	96.0
		3	32,553 ± 1582 ^a^	85.3	71,766 ± 4189 ^def^	99.0
		5	32,669 ± 1134 ^a^	85.7	65,640 ± 1565 ^bc^	90.5
	150	1	34,956 ± 880 ^abc^	91.7	67,996 ± 3566 ^cd^	93.8
		3	34,122 ± 1559 ^ab^	89.5	69,972 ± 3548 ^cdef^	96.5
		5	32,950 ± 1321 ^a^	86.4	63,099 ± 4955 ^ab^	87.0
	220	1	35,543 ± 1428 ^abc^	93.2	69,991 ± 3480 ^cdef^	96.5
		3	34,475 ± 1052 ^abc^	90.4	65,589 ± 4179 ^bc^	91.6
		5	34,591 ± 742 ^abc^	90.7	69,169 ± 2742 ^cde^	95.4
20	50	1	41,175 ± 1038 ^cde^	108.0	74,358 ± 3930 ^f^	102.6
		3	47,908 ± 1445 ^ef^	125.6	82,701 ± 3683 ^g^	114.1
		5	50,014 ± 981 ^g^	131.2	100,197 ± 3770 ^h^	138.2
	150	1	40,812 ± 1206 ^bcd^	107.0	73,085 ± 2434 ^ef^	100.8
		3	48,880 ± 2011 ^f^	128.2	79,306 ± 2513 ^g^	109.4
		5	45,239 ± 1420 ^def^	118.6	81,411 ± 2405 ^g^	112.3
	220	1	41,199 ± 619 ^cde^	108.1	65,203 ± 2717 ^bc^	89.9
		3	32,148 ± 404 ^a^	84.3	62,961 ± 4026 ^ab^	86.8
		5	33,741 ± 829 ^a^	88.5	59,224 ± 3367 ^a^	81.7

The values in a column with different superscript letters (a–h) are significantly different (*p* < 0.05).

## Data Availability

All data created and analyzed during the experiments was presented in this study.

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
