# Peer review of "Impact of High-Pressure Homogenization Parameters on Physicochemical Characteristics, Bioactive Compounds Content, and Antioxidant Capacity of Blackcurrant Juice"

_molecules, 2021, doi:10.3390/molecules26061802_

Round 1
Reviewer 1 Report
In this study, the authors investigated the effect of high-pressure homogenization (vs. pasteurization) on the quality of cloudy blackcurrant juice. They specifically looked into the effect of pressure, number of passes, and inlet temperature and measured titratable acidity, pH turbidity, color changes, anthocyanin, vitamin C and total phenolics content, and antioxidant activity.
In general, the discussion (intro and discussion of results) was thorough and in most cases well-articulated. Here are minor comments that the authors can address to improve the manuscript.
- References are missing in Line 53 and 137 where the authors refer to "literature" (i.e., According to the literature) but then missed citing the references.
- L34 - replace "exposition" with "exposure"
- The paragraph encompassing L143 to 148 needs proofreading.
- "There were found statistically..." does not sound correct.
- L146: "showed" instead of "shown"
- L148: "which" instead of "what"
- L159 - Did the author mean that the heat treatment did not "change" the proportion of the individual monomers, rather than "disturb"?
- L163-165 - The latter half of the sentence is vague.
- L171 - Instead of "shares", should be "distribution" or percentages
- L178 - Remove "substances"
- L181 - Rephrase into, "...acids, which are abundant in blackcurrant berries"
- L203 - Statement is vague
- L238 - "noted" instead of "notified"
- L248 - instead of saying "up or down" simply use ±5%
Author Response
- References are missing in Line 53 and 137 where the authors refer to "literature" (i.e., According to the literature) but then missed citing the references.
Authors: We have completed sentence from line 53 with the missing citations. The mentioned effect of the HPH process on the colour of the products in line 137 has citations of literature in the next sentence.
- L34 - replace "exposition" with "exposure"
- The paragraph encompassing L143 to 148 needs proofreading.
- "There were found statistically..." does not sound correct.
- L146: "showed" instead of "shown"
- L148: "which" instead of "what"
- L159 - Did the author mean that the heat treatment did not "change" the proportion of the individual monomers, rather than "disturb"?
- L163-165 - The latter half of the sentence is vague.
- L171 - Instead of "shares", should be "distribution" or percentages
- L178 - Remove "substances"
- L181 - Rephrase into, "...acids, which are abundant in blackcurrant berries"
- L203 - Statement is vague
- L238 - "noted" instead of "notified"
- L248 - instead of saying "up or down" simply use ±5%
Authors: All the above comments were valuable, they have been corrected like the whole article by language verification and proofreading with native English speaker.
Reviewer 2 Report
The using of high-pressure homogenization is very interesting topic in Food Science and Technology. The manuscript is well wrirtten, but deep revision is neccesseary:
First, authors should display the section Material and methods, than section of Results and discussion in the main manuscript;
It is an essential that authors include the Conclusions in the manin manuscript;
Did authors examine sensory and microbiological quality of the final products obtain using HPH? It is very important for consumers.
Author Response
First, authors should display the section Material and methods, than section of Results and discussion in the main manuscript;
Authors: According to the instructions for authors submitting an article to the journal Molecules and the template sent by the Editor, the Results and Discussion section should be placed before the Materials and Methods section.
It is an essential that authors include the Conclusions in the manin manuscript;
Authors: According to the instructions for authors submitting an article to the journal Molecules and the template sent by the Editor, the Conclusions section is not mandatory but can be added to the manuscript if the discussion is unusually long or complex. After further consideration, inspired by the reviewer comment, we decided to add a Conclusions section. Please find it in revised form of manuscript.
Did authors examine sensory and microbiological quality of the final products obtain using HPH? It is very important for consumers.
Authors: Microbiological analysis of the acidic products such as blackcurrant juices after HPH processing is the second stage of our research, including storage. Also sensory quality of these products combined with the study of consumer acceptance of products preserved by this innovative method will be a separate subject of our research. The results of the above researches will be published as articles in the near future.
Round 2
Reviewer 2 Report
I suggest to accept this manuscript.